# Human Endothelial Progenitor Cells Protect the Kidney against Ischemia-Reperfusion Injury via the NLRP3 Inflammasome in Mice

**DOI:** 10.3390/ijms23031546

**Published:** 2022-01-28

**Authors:** Ha Nee Jang, Jin Hyun Kim, Myeong Hee Jung, Taekil Tak, Jung Hwa Jung, Seunghye Lee, Sehyun Jung, Se-Ho Chang, Hyun-Jung Kim

**Affiliations:** 1Division of Nephrology, Department of Internal Medicine, Gyeongsang National University Hospital, Gyeongsang National University College of Medicine, Jinju 52727, Korea; asaku@naver.com (H.N.J.); bsbliz87@naver.com (S.L.); holyhyun@naver.com (S.J.); shchang@gnu.ac.kr (S.-H.C.); 2Institute of Health Sciences, Gyeongsang National University, Jinju 52727, Korea; ajini7044@hanmail.net (J.H.K.); jhring@hanmail.net (J.H.J.); 3Biomedical Research Institute, Gyeongsang National University Hospital, Jinju 52727, Korea; yallang7@hanmail.net; 4Gyeongnam Biohealth Research Center, Korea Institute of Toxicology, Jinju 52834, Korea; ttkil@kitox.re.kr; 5Division of Endocrinology, Department of Internal Medicine, Gyeongsang National University Hospital, Gyeongsang National University College of Medicine, Jinju 52727, Korea

**Keywords:** acute kidney injury, chronic kidney disease, endothelial progenitor cell, epithelial-mesenchymal transition, inflammasome, ischemia-reperfusion injury, fibrosis

## Abstract

Ischemia-reperfusion injury (IRI) is a major cause of acute kidney injury (AKI) and progression to chronic kidney disease (CKD). However, no effective therapeutic intervention has been established for ischemic AKI. Endothelial progenitor cells (EPCs) have major roles in the maintenance of vascular integrity and the repair of endothelial damage; they also serve as therapeutic agents in various kidney diseases. Thus, we examined whether EPCs have a renoprotective effect in an IRI mouse model. Mice were assigned to sham, EPC, IRI-only, and EPC-treated IRI groups. EPCs originating from human peripheral blood were cultured. The EPCs were administered 5 min before reperfusion, and all mice were killed 72 h after IRI. Blood urea nitrogen, serum creatinine, and tissue injury were significantly increased in IRI mice; EPCs significantly improved the manifestations of IRI. Apoptotic cell death and oxidative stress were significantly reduced in EPC-treated IRI mice. Administration of EPCs decreased the expression levels of NLRP3, cleaved caspase-1, p-NF-κB, and p-p38. Furthermore, the expression levels of F4/80, ICAM-1, RORγt, and IL-17RA were significantly reduced in EPC-treated IRI mice. Finally, the levels of EMT-associated factors (TGF-β, α-SMA, Snail, and Twist) were significantly reduced in EPC-treated IRI mice. This study shows that inflammasome-mediated inflammation accompanied by immune modulation and fibrosis is a potential target of EPCs as a treatment for IRI-induced AKI and the prevention of progression to CKD.

## 1. Introduction

Acute kidney injury (AKI) is a common comorbidity in critically ill patients and a major cause of death [1,2]. Ischemic-reperfusion injury (IRI) is the most common cause of AKI and progression to chronic kidney disease (CKD) [3]. Thus far, no effective therapeutic intervention has been established for AKI, and useful interventions require an understanding of the AKI to CKD transition.

The pathophysiology of ischemic AKI is a consequence of immunological and inflammatory processes accompanied by endothelial and epithelial cell injury [4,5]. In particular, renal inflammation is the main pathophysiology in ischemic AKI. IRI induces swelling, disrupts endothelial cells, and stimulates the inflammatory response. Experimental models suggest that the uncontrolled release of danger-associated molecular patterns (DAMPs) from damaged or dying cells drives the inflammatory response, as well as subsequent tissue and organ injury. Inflammasomes are multi-protein complexes that regulate the maturation of cytokines, inflammation, and cell death by activating specific caspases [6]. NLRP3 is the most well-characterized of the inflammasome-related proteins; it is activated by DAMPs, which regulate the secretion of proinflammatory cytokines such as interleukin (IL)-1β and IL-18. These inflammasome components have been directly implicated in renal inflammatory injury [7].

Endothelial progenitor cells (EPCs) are derived from the bone marrow or tissue-resident cells; they have major roles in the maintenance of vascular integrity and the repair of endothelial damage [8]. EPCs have function as therapeutic agents in various kidney diseases, particularly AKI [9]. Patschan et al. showed that EPCs have a renoprotective effect after IRI in mice. EPCs are mobilized after acute renal ischemia; they are recruited into ischemic kidney areas where they ameliorate AKI via paracrine effects and the repair of injured microvasculature [10]. Systemic injections of EPCs or their supernatant decrease renal dysfunction and interstitial fibrosis in ischemic AKI models [11]. The AKI recovery process and reduction of progression to CKD after AKI also involve EPC function. In this study, we examined whether human EPCs had renoprotective effects in an IRI-induced AKI mouse model.

## 2. Results

### 2.1. Culture and Differentiation of EPCs from PBMNCs

Several clusters, which is a typical feature of EPCs with spindle-shaped cells at the boundary, appeared within 3 days when peripheral blood mononuclear cells (PBMNCs) were cultured under endothelial cell culture conditions (Figure 1A). The morphologies of early and late EPCs were observed on days 7 and 14, respectively (Figure 1B,C). We evaluated EPC differentiation as the appearance of cells with a cobblestone morphology (i.e., late EPCs; Figure 1C) and the von Willebrand factor (vWF)-positive population was regarded as a typical endothelial feature (Figure 1D).

### 2.2. EPCs Ameliorate IRI-Induced Renal Dysfunction and Tissue Damage

Serum blood urea nitrogen and serum creatinine levels were significantly increased in the IR group. The EPC-treated IRI mice exhibited significant attenuation of the renal dysfunction associated with IRI (Figure 2A). Hematoxylin and eosin staining was performed to confirm IRI-induced tissue injury. Analysis of renal pathology in IRI mice revealed extensive tubular injury, characterized by tubular atrophy, cast formation, and brush border loss (Figure 2B). These pathological changes were significantly attenuated in EPC-treated IRI mice (Figure 2C).

### 2.3. EPCs Reduce IRI-Induced Apoptosis

Apoptosis contributes to the pathogenesis of IRI-induced AKI. The degree of apoptosis was assessed using a terminal deoxynucleotidyl transferase (TdT) dUTP nick-end labeling (TUNEL) assay. Apoptotic cell death was significantly reduced in tubular epithelial cells from EPC-treated IRI mice (Figure 3).

### 2.4. EPCs Attenuate IRI-Induced Oxidative Stress

To investigate the effects of EPCs on IRI-induced oxidative stress, immunohistochemical staining of 8-hydroxy-2′-deoxyguanosine (8-OHdG; a reactive oxygen species [ROS]-induced DNA damage marker) was performed using kidney tissue. 8-OHdG-positive signals were detected in the nuclei of tubular epithelial cells in the IRI-only group (arrow in Figure 4); these signals were significantly reduced in EPC-treated IRI mice (Figure 4).

### 2.5. EPCs Decrease Inflammasome Activation

We examined the expression patterns of inflammasome-related factors. A significant increase in NLRP-3 expression was observed in the kidneys of IRI mice; this increase was significantly attenuated in the EPC-treated IRI mice. Cleaved-caspase-1 (c-Casp-1), phosphorylated nuclear factor kappa B (p-NF-κB), and phosphorylated p38 (p-p38) were also ameliorated in the EPC-treated IRI mice (Figure 5A).

To confirm the effects of EPCs on inflammasome inactivation in IRI mice, we investigated the mRNA expression levels of IL-1β and IL-18, which are proinflammatory cytokines related to the NLRP3 inflammasome. IL-1β and IL-18 mRNA expression levels increased in the IRI-only mice and decreased in the EPC-treated IRI mice (Figure 5B).

### 2.6. EPCs Reduce Infiltration of Inflammation-Related Immune Cells

F4/80 immunohistochemical staining showed that EPC administration significantly reduced the infiltration of macrophages compared with the findings in IRI-only mice (Figure 6A). The expression level of intracellular adhesion molecule-1 (ICAM-1) was significantly reduced in EPC-treated IRI mice (Figure 6B). The mRNA expression levels of C-X3-C motif chemokine ligand 1 (CX3CL1) and the C-X3-C motif chemokine receptor (CX3CR1) were significantly reduced in the EPC-treated IRI mice (Figure 6C). The mRNA expression levels of the transcription factors RAR-related orphan receptor gamma T (RORγT) and IL-17RA were also significantly reduced in the EPC-treated IRI mice (Figure 6D).

### 2.7. EPCs Reduce IRI-Induced Renal Fibrosis via the EMT Pathway

Immunohistochemical staining of the myofibroblast marker α-SMA was performed to verify fibrosis. α-SMA-positive signals were found in the tubulointerstitial areas of the kidneys that underwent IRI. The number of positive signals was significantly reduced in the EPC-treated IRI mice (Figure 7A). The α-SMA mRNA expression level was also reduced (Figure 7B). The expression levels of transforming growth factor (TGF)-β1, Twist, and Snail mRNA (i.e., epithelial-mesenchymal transition [EMT]-specific biomarkers) were significantly reduced in the EPC-treated IRI mice (Figure 7B).

## 3. Discussion

This study showed that EPCs protected against IRI-induced histological and biochemical changes, as well as IRI-induced renal functional loss. In particular, EPCs inactivated the inflammasome-dependent NF-κB signaling pathway, decreased the infiltration of immune cells, and inhibited fibrosis.

Many studies have reported various pathophysiologies (e.g., apoptosis, inflammation, hypoxic injury, and oxidative stress) because of ROS production, which contributes to the pathogenesis of IRI-induced AKI [12,13,14]. The inhibition of these processes is advantageous when attempting to prevent and treat IRI-induced-AKI. Our experiments showed that the administration of EPCs significantly ameliorated IRI-induced TUNEL-positive apoptotic cell death, 8-OHdG-positive signals, infiltration of macrophages and Th17 cells, and activation of NF-κB expression. IRI begins with macrophage infiltration. Infiltrating macrophages generate ROS and produce proinflammatory cytokines in damaged tissues. ROS and proinflammatory cytokines activate the NF-κB signaling pathway, which promotes the transcription of adhesion molecules, such as ICAM-1. These adhesion molecules facilitate the migration of immune cells, such as Th17 cells. Ultimately, tubular epithelial cells undergo apoptotic cell death, and kidney dysfunction occurs. Taken together, these findings indicate that EPCs protect against IRI.

Macrophages exacerbate the AKI inflammatory response, as well as the associated cytotoxic effects, by generating ROS and proinflammatory cytokines [15]. The activation of NF-κB from ROS is important for disease progression; it promotes the synthesis of inflammatory mediators, leading to the transcription of adhesion molecules (e.g., ICAM-1) [16]. The inhibition of ICAM-1 expression reportedly decreased leukocyte adhesion and renal inflammation in an IRI model [17]. Collett et al. showed that conditioned medium from endothelial colony-forming cells preserves microvascular function in IRI by reducing ICAM-1 [18]. The attraction, retention, and differentiation of leukocytes are governed by chemokines in AKI. The chemokine CX3CL1 (fractalkine) and its receptor CX3CR1 are highly expressed and upregulated in AKI. The regulatory mechanisms of the CX3CL1–CX3CR1 axis are important for the localized and systemic effects of renal inflammation [19]. The CX3CL1–CX3CR1 interaction mediates macrophage–mesothelial crosstalk and promotes peritoneal fibrosis. Crosstalk between the CX3CL1–CX3CR1 axis and the NF-κB signaling pathway directly contributes to fructose-induced kidney injury; inhibition of the CX3CL1–CX3CR1 pathway suppresses renal-related diseases [20]. Th17 cells are mediators of inflammation in AKI and CKD. The differentiation and IL-17 activation of Th17 cells occur secondary to T-cell receptor activation. The transcription factor RORγT mediates the transcription of IL-17 [21]. Mehrotra et al. showed that renal protection in the RORγT knockout rat with IRI was associated with inhibition of IL-17 expression, as well as reduced infiltration of CD4+ cells, CD8+ cells, B cells, and macrophages [22]. The present study showed that the administration of EPCs reduced macrophage infiltration (measured by F4/80-positive signals) and Th17 cell infiltration (measured by RORγT and IL-17 receptor mRNA expression patterns). These changes could be related to the inactivation of NF-κB caused by reduced oxidative stress.

Inflammation is a major contributor to the pathophysiology of AKI. In particular, the NLRP3 inflammasome component is an important mediator of IRI [23,24]. The NLRP3 inflammasome is activated in AKI and CKD. The inhibition of NLRP3 (by NLRP3 inflammasome knockout or cathepsin-mediated NLRP3 inhibition) confers significant protection against IRI in mice [23,25]. Candidates that block NLRP3 inflammasome activation, such as hydroxychloroquine, are under investigation for the treatment of IRI-induced AKI [25]. IRI-induced activation of the NLRP3 inflammasome results in prolonged caspase-1 cleavage [24]. Similarly, caspase-1, a downstream target of NLRP3, has an important role in IRI [26]. To our knowledge, no study has examined whether EPCs inhibit activation of the NLRP3 inflammasome in IRI, although the NLRP3 inflammasome is an important mechanism in IRI. The present study found that EPCs attenuated the activation of IRI-induced NLRP3 inflammasome signaling, including NLRP3, cleaved caspase-1, NF-κB, and p-p38. Moreover, IL-1β and IL-18 mRNA levels, endpoints of NLRP3 inflammasome signaling, were significantly reduced in the EPC-treated IRI group. These data suggest that the use of EPCs and inactivation of the inflammasome has potential as a renal protection mechanism after IRI.

AKI is a risk factor for the development of CKD. Major causes for the AKI to CKD transition are incomplete or maladaptive repair after AKI and the onset of tubulointerstitial fibrosis [27]. The accumulation of macrophages promotes glomerular and interstitial fibrosis; it has a key role in renal inflammation [28]. Furthermore, there is evidence to support a TGF-β1-induced EMT [29]. Changes in the expression levels of TGF-β1, α-SMA, Twist, and Snail (i.e., EMT-specific biomarkers) are associated with NF-κB-mediated inflammation in the unilateral ureteric obstruction model [30,31]. In this study, EPCs exhibited fewer EMT-specific markers and less fibrosis in the IRI model.

Various cell types have been utilized for kidney injury treatment as therapeutic agents, including stem cells, progenitor cells, and primary cells derived from peripheral blood, cord blood, bone marrow, and adipose tissue, in experimental and clinical conditions. Most of advanced clinical trials are based on mesenchymal stem cells. It is less commonly applied to EPCs compared to other cells in kidney diseases. A few studies have reported that cell therapy using EPCs successfully treated patients with CKD [32] and AKI [33]. Although the current study did not show the human data, our results suggest that human peripheral blood-derived EPCs might be a promising therapeutic option to treat kidney diseases through the inactivation of the NLRP3 inflammasome.

In conclusion, the pathogenesis of renal IRI contains multiple complex steps and mechanisms. Therefore, targeting a single step and a single mechanism is not helpful for the treatment of IRI-induced AKI. To our knowledge, this is the first report to indicate that EPCs could be a potential option for multiple targets, including the suppression of inflammasome-mediated inflammation and fibrosis as a preventive approach for ischemic AKI. Our findings suggest that inflammasome-mediated inflammation, accompanied by immune modulation and fibrosis, constitute a potential target of EPCs for the treatment of IRI-induced AKI and the prevention of progression to CKD.

## 4. Materials and Methods

### 4.1. Ethics Statement

This study was approved by the Gyeongsang National University Institutional Animal Care (GNU160216-M0009) for the animal study and by the Gyeongsang National University Hospital Institutional Ethics Committee (GNUH 2016-05-003-002) for the human study.

### 4.2. Isolation and Culture of EPCs from Human PBMNCs

All blood samples were processed within 2 h after blood collection. PBMNCs were isolated by Ficoll density gradient centrifugation (Sigma, St. Louis, MO, USA) for 25 min at 2300 rpm, followed by three washes in phosphate-buffered saline. The cells were plated on culture dishes coated with human fibronectin (Sigma), then cultured in endothelial cell growth medium (EGM-2; Clontech, Mountain View, CA, USA) containing EGM SingleQuots. After 3 days, nonadherent cells were removed, and a fresh culture medium was applied. Cells (1 × 10^5^) on day 14 were vWF-stained. All cell cultures were maintained at 37 °C with 5% CO_2_ in a humidified atmosphere.

### 4.3. IRI-Induced AKI

Ten-week-old male C57BL/6 mice were maintained under a 12-h/12-h light/dark cycle in a temperature- and humidity-controlled facility. Standard mice chow and water were provided ad libitum. Mice were assigned to the sham (Sham), EPC, IRI-only (IR), and IRI with EPC (IR + EPC) groups (*n* = 10 per group).

Mice were anesthetized with an intraperitoneal injection of Avertin (2,2,2-tribromoethanol, Sigma). The renal pedicles were bilaterally clamped with microaneurysm clamps for 40 min after a midline incision. Ischemia time was chosen to obtain a reversible model of ischemic AKI and to avoid mortality. EPCs (5 × 10^5^ cells, tail vein) were administered 5 min before reperfusion. After clamp removal, the kidneys were observed for blood flow restoration and return to their original color. The abdomen was closed in two layers. The sham surgery consisted of the same surgical procedure without the application of clamps. During the first 24 h of reperfusion, the animals were held in an incubator at 29 °C. The animals were sacrificed 72 h after ischemia. Blood and kidney tissues were harvested. All experiments were performed in triplicate.

### 4.4. Histopathology

Tissues were fixed in 4% paraformaldehyde in 0.1 M phosphate-buffered saline, embedded in paraffin, and cut into 5-μm sections. The sections were stained with hematoxylin and eosin. Semiquantitative scoring for hematoxylin and eosin staining was based on the degree of interstitial injury assigned in points (0–3) for the extent of interstitial fibrosis and tubular atrophy (defined as luminal dilation, brush border loss, and flattened tubular epithelial cells). Tissue injury was scored by grading the percentage of affected cells under a high-powered field (×400): 0, 0%; 1, <30%; 2, 31–60%; 3, 61–100%. All scores were summed and represented as mean values on a graph; the signals were analyzed using NIS-Elements BR 3.2 (Nikon, Tokyo, Japan).

### 4.5. TUNEL Assay

The degree of apoptosis was assessed using a TUNEL assay. DNA fragmentation was detected using a kit from Roche Applied Sciences (Indianapolis, IN, USA). Semiquantitative analysis was performed by counting the number of TUNEL-positive cells per field in the renal tissue at ×400 magnification. At least 10 areas were randomly selected in the cortex per slide. The mean number of brown-colored cells in the selected fields was expressed as the density of TUNEL-positive cells.

### 4.6. Immunoblotting

Kidney samples were obtained for immunoblotting. The tissues were homogenized in RIPA buffer (Thermo Scientific, Waltham, MA, USA). The amount of protein was measured using a BCA assay kit (Pierce, Rockford, IL, USA) in accordance with the manufacturer’s protocol. Protein samples (50 µg) were subjected to sodium dodecyl sulfate-polyacrylamide gel electrophoresis and transferred to membranes for blotting. The blots were probed with monoclonal primary antibodies against NLRP-3 (Abcam, Cambridge, UK), c-Casp-1 (Abcam), p-NF-κB (Santa Cruz Biotechnology, Santa Cruz, CA, USA), or p-p38 (Cell Signaling Technology, Danvers, MA, USA) at 4 °C overnight. Primary antibody binding was visualized with a secondary antibody and an ECL kit (Amersham Pharmacia Biotech, Piscataway, NJ, USA). The β-actin antibody (Sigma) served as the loading control. Densitometric analysis was performed to quantitatively analyze the data.

### 4.7. Immunohistochemistry

After deparaffinization, the sections were incubated with monoclonal antibodies against ICAM-1 (BD Bioscience, Franklin Lakes, NJ, USA) or α-SMA (Sigma), or polyclonal antibodies against F4/80 (Santa Cruz Biotechnology) or 8-OHdG (Abcam); they were then incubated with biotin-conjugated secondary IgG (diluted 1:200; Vector Laboratories, Burlingame, CA, USA), avidin-biotin-peroxidase complex (ABC Elite Kit; Vector Laboratories), and DAB. Next, sections were visualized by light microscopy; digital images were captured and analyzed using NIS-Elements BR 3.2. Semiquantitative analysis was performed by counting the number of immunohistochemically stained positive cells per field in the renal tissue at ×400 magnification.

### 4.8. Quantitative Real-Time-Polymerase Chain Reaction (PCR)

Kidney samples were obtained for quantitative real-time PCR. Total RNA was isolated from frozen kidney tissues using TRIzol (Invitrogen, Carlsbad, CA, USA). The purified RNA was reverse transcribed into cDNA using an iScript cDNA synthesis kit (Bio-Rad Laboratories, Hercules, CA, USA). Quantitative cDNA amplification was performed using the ViiA7 Real-Time System (Applied Biosystems Inc., Foster City, CA, USA), Power SYBR Green PCR Master Mix (Applied Biosystems), and gene-specific primers for IL-1β, IL-18, CX3CL1, CX3CR1, RORγt, IL-17RA, TGF-1, α-SMA, Twist, and Snail. GAPDH was used as the internal control to normalize the quantity of RNA. The relative gene expression level in each sample was quantified using the 2DDCt method. The following primer sequences were used: IL-1β, CTTCAGGCAGGCAGTATCACTCAT (F), TCTAATGGGAACGTCACACACCAG (R); IL-18, GCTGTGACCCTCTCTGTGAA (F), GGCAAGCAAGAAAGTGTCCT (R); CX3CL1, ATTGGAAGACCTTGCTTTGG (F), GCCTCGGAAGTTGAGAGAGA (R); CX3CR1, CACCATTAGTCTGGGCGTCT (F), GATGCGGAAGTAGCAAAAGC (R); RORγt, AGCTTTGTGCAGATCTAAGG (F), TGTCCTCCTCAGTAGGGTAG (R); TGF-1, TGCGCTTGCAGAGATTAAAA (F), CGTCAAAAGACAGCCACTCA (R); Twist, CTCGGACAAGCTGAGCAAG (F), CAGCTTGCCATCTTGGAGTC (R); Snail, CTTGTGTCTGCACGACCTGT (F), CTTCACATCCGAGTGGGTTT (R); GAPDH, ACTCCACTCACGGCAAATTC (F), TCTCCATGGTGGTGAAGACA (R); IL-17RA (Taqman, Mm.00434214); and α-SMA (Taqman, Mm.00725412).

### 4.9. Statistical Analysis

Statistical analyses were performed using GraphPad Prism software (version 8.0; GraphPad Software Inc., La Jolla, CA, USA). Data were evaluated using a one-way analysis of variance and Tukey’s multiple comparison test (to compare all groups). A *p*-value < 0.05 was considered statistically significant.

## Figures and Tables

**Figure 1 ijms-23-01546-f001:**
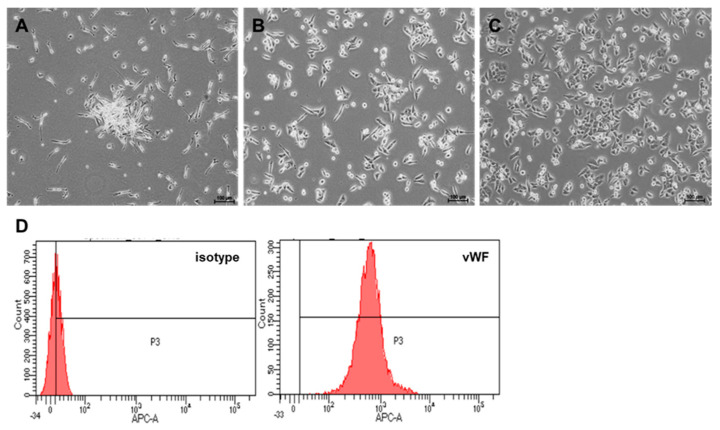
In vitro culture and differentiation of EPCs from PBMNCs. Representative photomicro-graphs of a cell cluster on day 3 (**A**), early EPCs on day 7 (**B**), and late EPCs developed on day 14 (**C**) from healthy volunteers. Expression of the endothelial marker vWF from late EPCs (**D**). The corresponding negative isotype control is shown on the left (isotype) (**D**).

**Figure 2 ijms-23-01546-f002:**
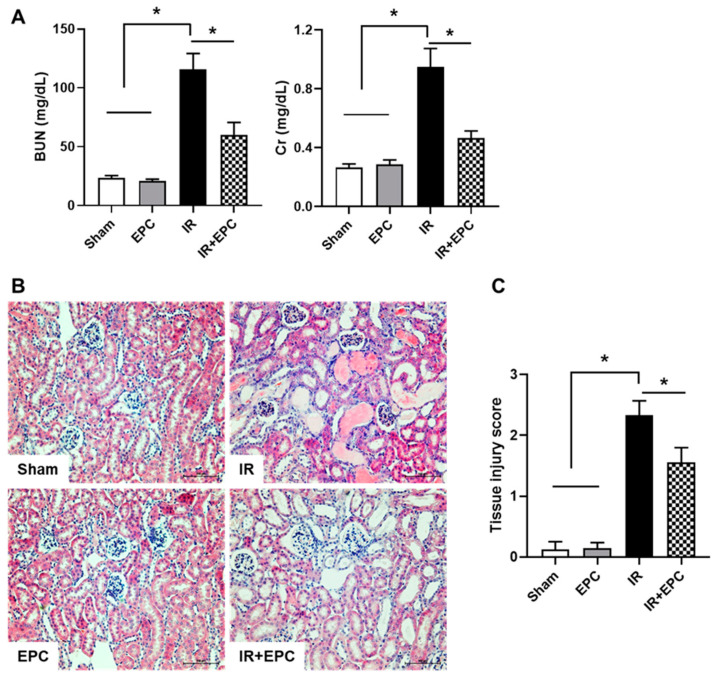
Effects of EPCs on renal function and morphological changes during IRI. EPCs (5 × 10^5^ cells) were administered through the tail vein. Mice were sacrificed 72 h after ischemic injury for blood and kidney sampling. The blood urea nitrogen (BUN) and serum creatinine (Cr) levels were measured (**A**). Histological changes were examined by hematoxylin and eosin staining (**B**), and tissue damage was quantified (**C**). Scale bar, 100 μm. Data are means ± SEMs. * *p* < 0.05.

**Figure 3 ijms-23-01546-f003:**
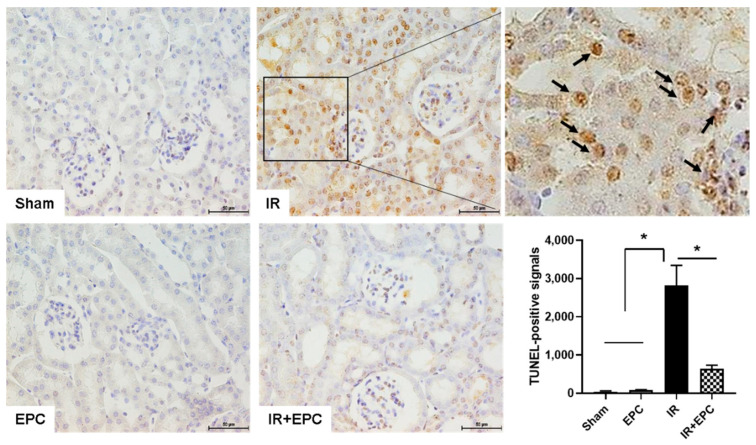
Effects of EPCs on IRI-induced apoptosis. Apoptotic cell death was examined by the TUNEL assay. Quantitative analysis of TUNEL-positive cells was performed. Scale bar, 50 μm. Data are means ± SEMs. * *p* < 0.05.

**Figure 4 ijms-23-01546-f004:**
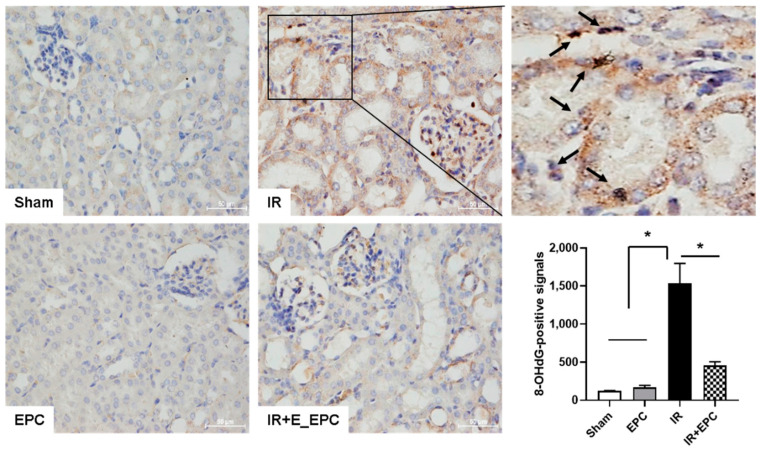
Effect of EPCs on IRI-induced oxidative stress. Sections were stained with anti-8-OHdG as an oxidative stress marker. Signals were analyzed by densitometry. Scale bar, 50 μm. Data are means ± SEMs. * *p* < 0.05.

**Figure 5 ijms-23-01546-f005:**
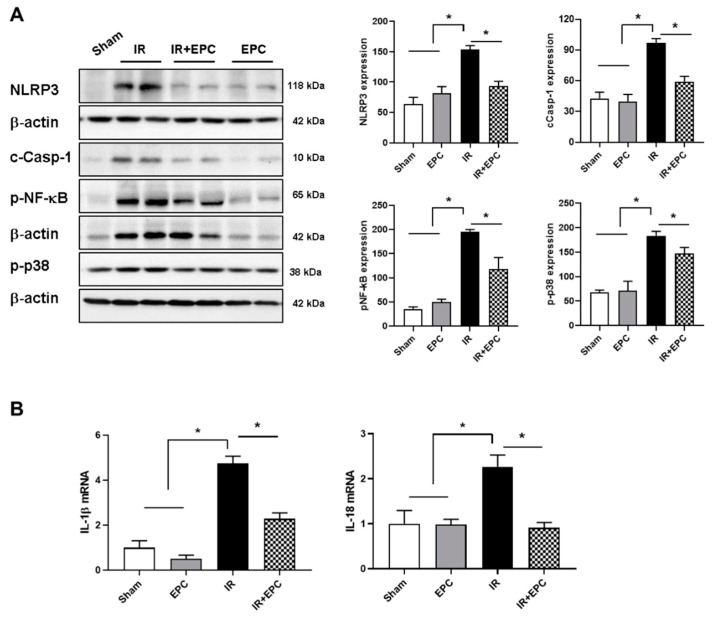
Effects of EPCs on the expression patterns of IRI-induced inflammasomes. A renal extract was prepared 72 h after IRI. Expression levels of NLRP-3, c-Casp-1, p-NF-κB, and p-p38 (**A**) were analyzed by immunoblotting. Quantitative analyses of NLRP-3, c-Casp-1, p-NF-κB, and p-p38 were performed, and the results were normalized to the levels of β-actin. mRNA expression levels of IL-1β and IL-18 were measured by quantitative real-time PCR (**B**). Data are means ± SEMs. * *p* < 0.05.

**Figure 6 ijms-23-01546-f006:**
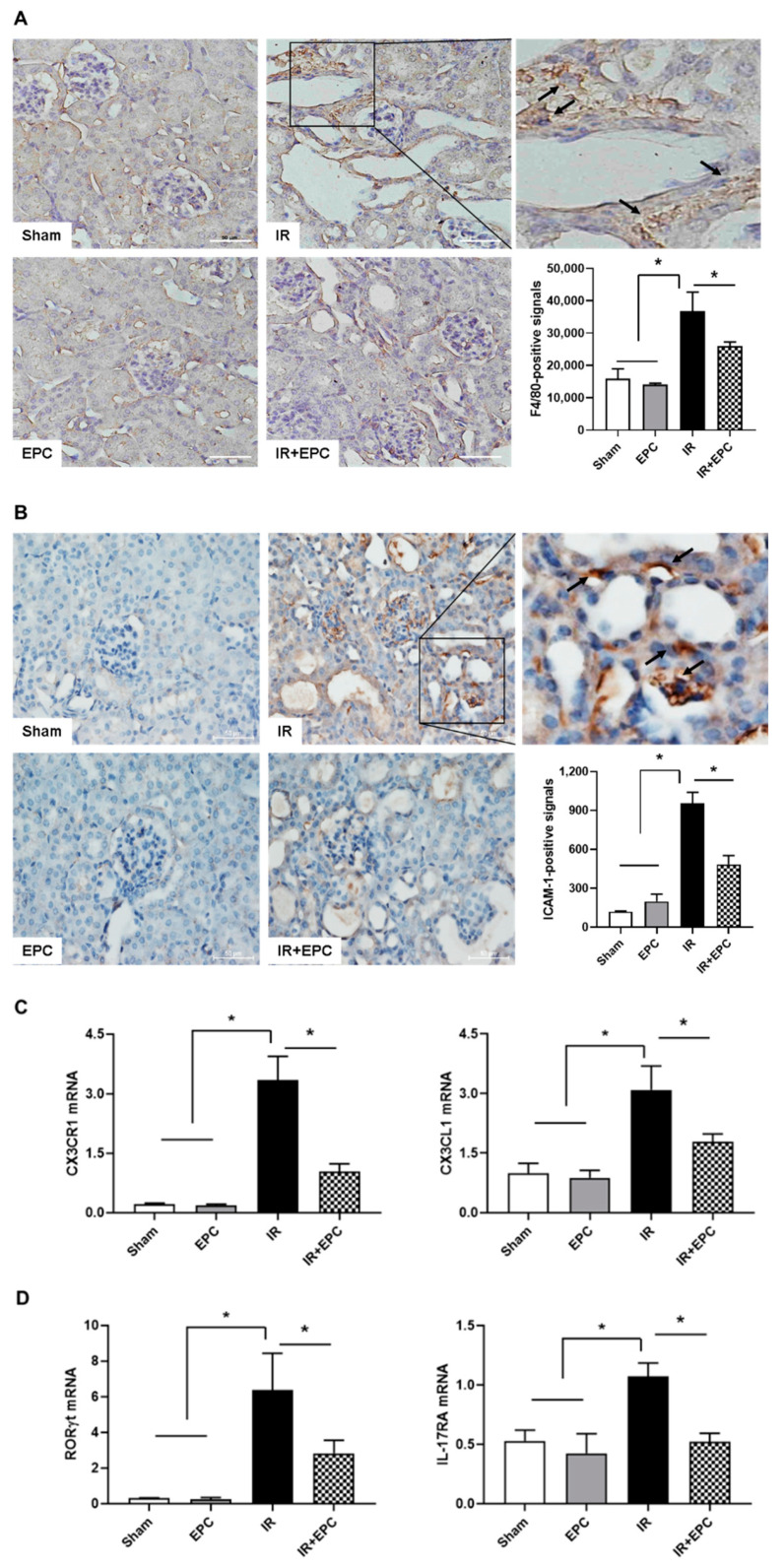
Effects of EPCs on IRI-induced inflammation by immune modulation. F4/80 immunohistochemical staining was performed to verify macrophage infiltration (**A**). F4/80-positive signals were found in the interstitial areas of the kidneys after IRI. Immunohistochemical staining of the inflammatory mediator ICAM-1 was examined (**B**). All signals were analyzed by densitometry. mRNA expression levels of CX3CL1, CX3CR1, RORγt, and IL-17RA were analyzed by quantitative real-time PCR (**C**,**D**). Scale bar, 50 μm. Data are means ± SEMs. * *p* < 0.05.

**Figure 7 ijms-23-01546-f007:**
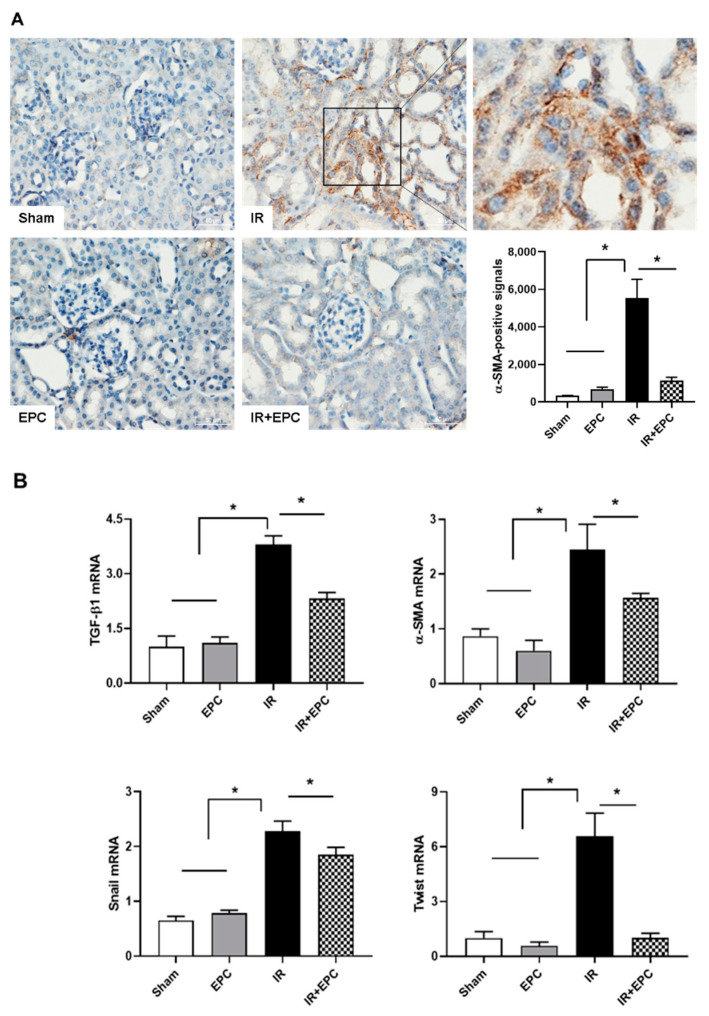
Effects of EPCs on IRI-induced renal fibrosis. α-SMA immunohistochemical staining was performed to verify fibrosis (**A**). mRNA expression levels of TGF-β1, α-SMA, Twist, and Snail were analyzed by quantitative real-time PCR (**B**). Scale bar, 50 μm. Data are means ± SEMs. * *p* < 0.05.

## Data Availability

Not applicable.

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
