# Peer review of "Human Endothelial Progenitor Cells Protect the Kidney against Ischemia-Reperfusion Injury via the NLRP3 Inflammasome in Mice"

_ijms, 2022, doi:10.3390/ijms23031546_

Round 1

Reviewer 1 Report

The authors established the renoprotective role of Endothelial progenitor cells (EPCs) in a mouse model for Ischemia-reperfusion injury (IRI). They demonstrated that the administration of EPCs of human origin reduces cell death and oxidative stress in IRI mice. Additionally, they observe that inflammation was reduced as well when mice were treated with EPCs.

Although the importance of EPCs to ameliorate kidney injury is not novel, the authors were very thorough with their investigation of inflammation specifically, and therefore these high-quality results offer a solid base to future investigators analyzing this problem.

Minor comments:

Line 79-figure legend: (D) should be bold.

Major comments:

In figure 5 NLPR3 signal in IR+EPC is not very clear: the first sample has an incomplete band, the second sample has a level comparable to the IR samples. Bands in EPC samples are also not clearly visible (maybe this is due to some air bubbles during transfer). If this experiment was performed more than once a better blot should be shown, where bands are clearly visible in all the samples (if samples are still available another blot could be run). This is a very central finding of this paper therefore is important to have a clear blot.

Please add kDa sizes on the side of the blot. If more than one blot was run, show the beta-actin control for every single experiment (if all the immunoblots were stained from a single membrane this is not necessary).

Line 266, statement: ”all experiments were performed in triplicate”. Does this refer to the following experiments (histopathology, immunoblotting, and so on) or to the IRI-induced AKI? The authors previously state that there are 10 mice per group. Please clarify how many times each set of experiments was performed.

Reviewer 2 Report

The article would assess mechanistic insights in renal failure associated with ischemia/reperfusion injury. The article focuses on in vitro and in vivo experiments targeting tthe role of Endothelial progenitor cells and misses translation into the clinical setting. I would suggest authors to explore this aspects using an already collected cohort of human beings (if available) or better discuss clinical relevance of the discovery in the discussion.

Round 2

Reviewer 1 Report

I believe the manuscript has been sufficiently improved to warrant publication in IJMS.